# Multi-Function Tradeoffs of Land System in Urbanized Areas—A Case Study of Xi'an, China

**Jiaqi Shao** [1] **and Fei Li** [1,2,3]*

1   College of Urban and Environmental Sciences, Northwest University, Xi'an 710127, China; shaojiaqi@stumail.nwu.edu.cn
2   Yellow River Institute of Shaanxi Province, Xi'an 710127, China
3   Shaanxi Key Laboratory of Earth Surface System and Environmental Carrying Capacity, Xi'an 710127, China
*   Correspondence: lifei@nwu.edu.cn

**Abstract:** Multi-functional trade-offs and synergy research on land systems are hotspots in geography and land science research, and are of great significance for achieving sustainable development of land use and the effective allocation of land resources. Recently, the development of the western region and The Belt and Road Initiative have become key topics, bringing opportunities and challenges to Xi'an. The rapid development of cities is accompanied by drastic changes in land use, and the ecological problems in the Qinling Mountains are becoming increasingly severe. This study took Xi'an as a case study and quantitatively evaluated the spatial-temporal patterns and trade-offs of land system functions such as economic development (*ED*), grain production (*GP*), ecological service (*ES*), etc. on the scale of 1 km × 1 km by fusing the data on land use, topography, soil, climate, and social economy. The results showed that the *ED* function of the land system continued to rise between 1980 and 2015, the *GP* function first declined and then increased; however, the *ES* function continued to decline. The *ED*, *GP* and *ES* functions respectively present a spatial pattern of high-value agglomeration, high in the north and low in the south, and high in the south and low in the north. In general, the three land system functions were trade-offs between each other. In terms of spatial pattern, *ED* and *ES* functions showed trade-offs in the south and a synergy distribution in the north; *ES*s and *GP* function trade-off zone significantly larger than the synergy zone, the trade-off between the two was significant; while the trade-off and the synergy zone for *GP* and *ED* was relatively small, the trade-off zone was the main one. The significant trade-off between *GP* and *ES* functions of the land system is a serious problem in land use in Xi'an. Under the premise of limited arable land, it is the current feasible strategy to promote the high-quality development of agriculture to increase the cultivation rate and efficiency, and to strengthen the ecological protection of arable land. In addition, the continued decline of *ES* functions is also worthy of attention. It is necessary to focus on increasing the greening rate of the city and strengthening the ecological management of the northern foot of the Qinling Mountains.

**Keywords:** land system; Multifunction; trade-offs and synergy; Xi'an; space-time evolution

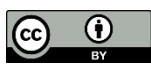

## 1. Introduction

As a natural and socio-economic complex, land has many functions and provides a variety of services for human society. It is an indispensable material basis for human production and life. The ability of land to provide products and services; that is, land use has multi-functional characteristics [1]. There is a relationship of mutual promotion and the trade-off between the functions of the land system, which has an important impact on the development and evolution of the land system [2]. The study of trade-off in the land system is not only a hot topic in geography and land science, but also provides new ideas for rational allocation of land resources and alleviating land use contradictions. Since the reform and opening up, the process of urbanization in China has been advancing rapidly. Due to the high degree of overlap in the spatial pattern of urban areas and cultivated land,

newly-added construction land is mainly converted from cultivated land. At the same time, in order not to break the red line of cultivated land and ensure grain security, a large amount of forest land has been cultivated to supplement cultivated land [3,4]. The disappearance of natural vegetation has led to the emergence of urban heat island effects, air pollution, and a lack of water resources in the process of urbanization, which has profoundly affected the livelihood and well-being of residents in urbanized areas. Therefore, our country's ecological civilization construction and sustainable development urgently need to coordinate the expansion of urban development space, the shrinking of natural ecological space, and the reconstruction of agricultural production space. The competition between urban development space, natural ecological space, and agricultural production space reflects the multi-functional game and conflict of the land system—namely, the trade-off between economic development (*ED*) function, grain production (*GP*) function, and ecological service (*ES*) function.

The study of land use multi-functionality evolved from the early agricultural multi-functionality [5]. Raised by The Sixth Framework Programme for "Research Sustainability Impact Assessment: Environmental, Social and Economic Effects of Multifunctional Land Use in Europe", formally defined as *Various products and services that can be provided to humans in the process of land use in a certain area* [5,6]. Later, foreign researchers carried out extensive research using the theory and methods of land use multifunctionality, and developed from land use pattern change to land space multifunctional change and its sustainability [7,8], which greatly expanded the connotation of land multi-functionality research: De Groot [9] combined function analysis and value evaluation to put forward a comprehensive evaluation framework of landscape ecology, economy, and social benefits, and diagnosed the conflicts of land functions to realize the sustainable use of landscape; Reidsma [10] and others took the agricultural water pollution problem as an example, assessed the impact of land use on the sustainable development of developing countries, and regarded the sustainable use and development of land as the ultimate goal of multifunctional research on land use; Schößer [11] passed the applicability of the three methods of landscape multi-functions, ecological multi-functions, and land system multi-functions which are compared, and it is pointed out that the three methods all aim at sustainable land use.

Among domestic researchers, Wang Chao (1984) first discussed the function of land, thinking that the function of land is certain traits and abilities manifested in the process of the internal and external connections of the land and the interaction with human activities, including the social and economic use function and its ability to maintain its own balance and stability [12]. Early research on multi-functionality focused on agriculture and rural areas. For example, Song Xiaoqing et al. [13] explored the possibility of arable land protection in our country from the perspective of arable land multi-functionality, and proposed that the research on the multi-functionality of agriculture is the basis of the multi-functional research of land use; Yuan Hong [14] et al. studied the use of multi-functional agricultural land in the process of urbanization to explore the best way to improve the living standards of farmers. In recent years, the research on land multifunctional utilization has gradually attracted some domestic researchers' attention [15–18], and a rich classification system has been established: most researchers divide it into social functions, economic functions, and ecological functions. Chen Jing et al. [19], on the basis of the existing classification system, further discussed the classification of land use functions, and established a land use function classification system that serves production, ecology, and living functions. In addition, Wang Feng and other researchers [20,21] broadened the composition of functions, introduced the function of culture, and proposed four major functions: economy, environment, society, and culture. Chen Ying et al. [22] extended the functions according to the local specific socio-economic and natural environmental characteristics, put forward six major functions such as raw material supply, carrying, transportation, ecological environment, landscape cultural heritage, and biological production function; Tao [23] and others have expanded production functions, supply functions, and

safety assurance functions. The above research adjusts and enriches the concept of the multi-functionality of the land system, and promotes the theoretical research of the multi-functionality of the land system. However, the evaluation of each function adopts the method of establishing an index system, and the evaluation mode of each function is carried out by assigning different weights to the index layer, affected by subjective factors.

In addition to the above theoretical research, the dimension of multi-source data dimensions is also the key to accurately evaluate the multi-function of the land system. At present, the commonly used methods of dimensional unification mainly include normalization method, area method, and monetization method. The normalization method is also called the standardization method, which avoids the differences in the comprehensive evaluation of various indicators [20]. However, horizontal comparison between different indicators is difficult to carry out; Xie Gaodi et al. measured the land function by the proportion of each land use area to the total land area [24]; the monetization method is to convert the products and services provided by the land system into a monetary dimension to facilitate intuitive comparison between different indicators [25]. On the research scale, the usual geographic research is mostly based on provinces, cities, districts, and counties to facilitate the acquisition of socio-economic data. Zhen Lin [26] evaluated the multi-functionality of land use across the country based on the data of various provinces. Taking the administrative area as the evaluation unit is convenient for the qualitative evaluation of the land use function under this scale, but there is a lack of precise research [5]. For this reason, some researchers try to use the area of different land use types as the data basis for function measurement, or use GIS technology to divide the regional land space into regular grids to achieve functional spatialization research.

To summarize, the current land system multi-function research has made great progress and fruitful results, but there are also the following problems: (1) It is difficult to reflect the true state of land system functions by selecting multi-layer indicators to indirectly calculate the value of them. In addition, the selection of indicators is greatly affected by subjective factors, and the evaluation results can't be compared horizontally in different regions. (2) Taking human beings as the demand side and land as the supply side, studied the potential ability of land system to provide products and services; (3) The evaluation is based on the administrative region as the unit, the regional differences are difficult to reflect. The results can only reflect the macro-level situation, so we need more micro-scale research results; (4) The relationship between the various functions of the land system is not clear, so we need to learn from the research framework of ecosystem services trade-off in the field of ecology to explore the pattern, process, and mechanism of the trade-off between the functions of the land system.

Therefore, this study is based on the perspective of supply and establishes a direct calculation model of the value of each function of the land system to reflect the true level and spatial pattern of each function of the land system, and then learn from the research methods of ecosystem service trade-offs to explore the space-time trade-off relationship of land system multi-function. The study not only explained the changes in the pattern of various functions of the land system in Xi'an and their trade-offs, and provided policy support for land use planning in Xi'an, but also designed to provide a reference for solving land use contradictions in rapidly urbanizing areas in developing countries. We also need to provide suggestions for rectification of land use contradictions caused by rapid urbanization at this stage, the weakening of ecological services caused by the disorderly expansion of cities, and food security.

## 2. Materials and Methods

### 2.1. Study Sites

The study area is located in Xi'an, in the central part of the Guanzhong Plain (Figure 1), bordered by the Weihe River in the north and Qinling Mountains in the south. The total area is 10,108 square kilometers. At the end of 2019, the permanent population was 10,203,500, of which the urban population was 7,612,800, and the urbanization rate was 74.61%. As shown in Figure 1, the land use types mainly include forest land, grassland, construction land, and cultivated land. Among them, the southern part is the Qinling Mountains, mostly forest land; the cultivated land and construction land are mostly distributed in northern Xi'an. The total cultivated land area is 332,000 hm$^2$, and the per capita cultivated land is 0.05 hm$^2$, which is lower than the national per cultivated land area and per capita cultivated land area. The contradiction of insufficient land resources is very prominent. In recent years, the implementation of the Western Development Strategy has accelerated the pace of urbanization in Xi'an. The rapid expansion of construction land has led to a reduction in the area of cultivated land. In order not to break the red line of cultivated land, grassland and woodland have also been cultivated to ensure grain security. Xi'an is in a period of rapid urbanization. There are problems such as environmental pollution and insufficient ecological land in the city, and the phenomenon of heat island is serious. Also, under the opportunities of the Western Development and the " The Belt and Road", Xi'an has ushered in rapid development, which puts forward higher requirements for the multi-functional land use [27]. Therefore, the research on land multi-function trade-off has extremely important theoretical and practical significance for the optimization of land resources and the sustainable use of regional land in Xi'an [28].

In the past, urbanization studies were mainly concentrated in the eastern coastal cities, the Yangtze River Delta, and the Beijing–Tianjin–Hebei area [29–31], on the other hand, has a pivotal position in the northwest region. It has both the universality and particularity of cities in the western region. First of all, in terms of development, under the opportunity of the "Belt and Road", Xi'an, as the starting point of the Silk Road, has the geographical advantage of connecting the east and the west. It has an important strategic position in the western development strategy. It is in a period of rapid urbanization; in terms of natural conditions, it belongs to a more common temperate continental monsoon climate with four distinct seasons. There is a huge difference in elevation within the territory, and the boundary between the Qinling Mountains and the Weihe Plain is clear, forming the main landform of Xi'an. In addition, the types of land use are rich and diverse, and they are concentrated and contiguous, which is conducive to the study of land use; in terms of ecological environment, Xi'an has been known as "Eight Waters Around Chang'an" since ancient times, and its water systems are abundant. In recent years, ecological governance problems in the Qinling Mountains have caused problems. Since the government attaches great importance to it, the study of land use in the Qinling Mountains cannot be delayed.

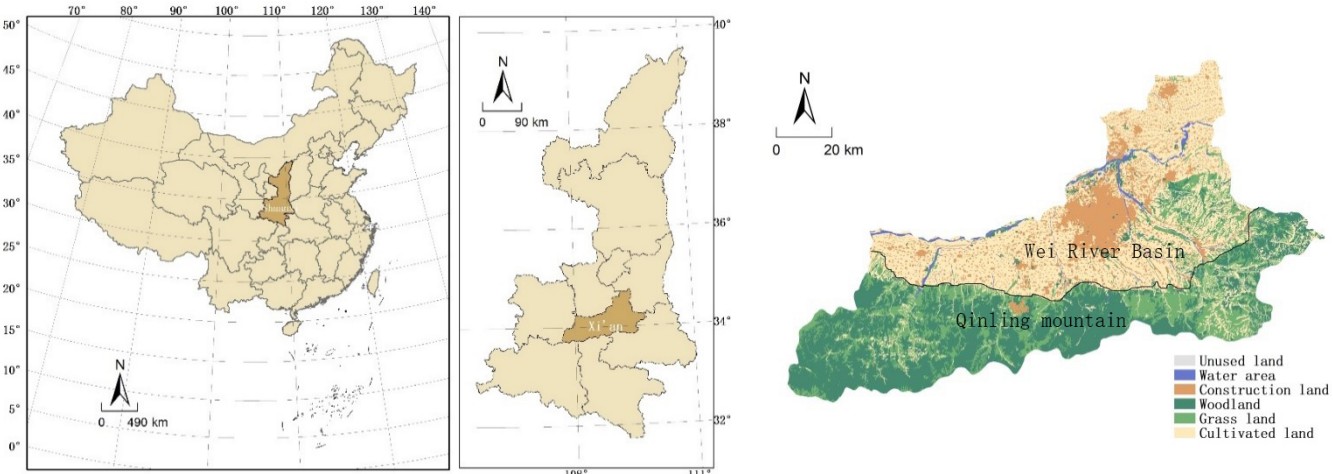

**Figure 1.** Xi'an location and land use map.

### 2.2. Data Sources

The data required for this study include land use data, climate data, soil data, topographic data, remote sensing image data, and socioeconomic statistics.

Land use data requires 3 periods (1980/2000/2015), mainly obtained from the second national land use survey database and the 1:100,000 land use database of the Resource and Environment Data Center of the Chinese Academy of Sciences. The climate data is downloaded from the China Meteorological Data Network (http://data.cma.cn/site/index.html (accessed on 2020/05/30)), including the monthly rainfall, average maximum temperature, average minimum temperature, average wind speed, average relative humidity, sunshine hours, rainfall days, sunshine percentage, total radiation, etc. The annual average rainfall and average temperature data from 1980 to 2015 are from the "Shaanxi Yearbook".

The soil data comes from the 1:1 million national soil data set of the Resource and Environmental Science Data Center of the Chinese Academy of Sciences, including soil type, soil composition, soil depth, and soil water holding capacity.

The terrain data comes from the Digital Elevation Model (DEM) data provided by the Shuttle Radar Topography Mission (SRTM) system. The spatial resolution of the SRTM-DEM used in this study is 90 m.

The socio-economic statistics are derived from the "Xi'an Yearbook" in the China Economic and Social Development Statistics Database, including six issues (1980/1990/2000/2005/2010/2015) of Xi'an GDP data.

### 2.3. Evaluation of Land System Function Value

This study uses ArcGIS as a platform to divide the study area into several grids of 1 km × 1 km, and calculate the value of various land system functions on each grid. Research based on the grid scale is convenient for spatial expression of land system functions, reveals their temporal and spatial distribution rules, and further analyzes the evolution characteristics of each function from 1980 to 2015.

#### 2.3.1. Functional Value Evaluation of   Economic Development (*ED*)

The *ED* function of the land system mainly considers the driving effect of the expansion of construction land on the gross national product. The calculation formula for its quantitative evaluation is as follows:

$$ED = \sum_{i=1}^{3} A_i \times Q_i \tag{1}$$

Among them, *ED* is the value of the *ED* function of construction land; $A_1$, $A_2$, and $A_3$ are the area of urban land, rural residential area and other construction land, respectively; $Q_1$, $Q_2$, and $Q_3$ are the *ED* function value corresponding to various construction land. Calculated by the following formula:

$$minf = \sum_{j=1}^{v} [\Delta GDP_j - \sum_{i=1}^{3} \Delta A_{ij} \times Q_t]^2 \tag{2}$$

$$
\begin{aligned}
Q1 &> 0 \\
Q2 &> 0 \\
Q3 &> 0 \\
Q1 - Q3 &> 0 \\
Q1 - Q3 &> 0 \\
Q3 - Q2 &> 0
\end{aligned}
\tag{3}
$$

Among them, j represents the corresponding time period (I: 1980–1990; II: 1990–2000; III: 2000–2005; IV: 2005–2010; V: 2010–2015); $\Delta GDP_j$ refers to the amount of GDP change during the corresponding time period; $\Delta A_{ij}$ refers to the area change of a certain type of construction land during that time period.

In order to express *ED* spatially, the study area is divided into a number of 1 km × 1 km grids, and the value of the *ED* function of the land system in each grid can be calculated separately according to Formula (1).

2.3.2. Functional Value Evaluation of Grain Production (*GP*)

The value of *GP* function (*GP*) of the land system is calculated by the following formula:

$$GP = \sum_{i=1}^{n} GrPP_i \times P_i \tag{4}$$

$GrPP_i$ refers to the multi-year average rainfed production potential of a certain type of grain crop, and $P_i$ is the unit price of the crop. Since the grain crops in Xi'an are mainly corn and wheat, which account for more than 90% of the total grain output, this study focuses on the rain-fed production potential of corn and wheat. The rainfed production potential of corn and wheat is estimated based on the use of the Global Agro-ecological Zone (GAEZ) model.

First, according to meteorological data such as precipitation, maximum temperature, minimum temperature, and solar radiation, using GAEZ model (Figure 2) to simulate the rain-fed production potential of corn and wheat in the study area in 1980/2000/2015; then, the average corn and wheat rainfed production potentials of the three years are calculated separately, and the value of the land system's *GP* function can be calculated using formula (4).

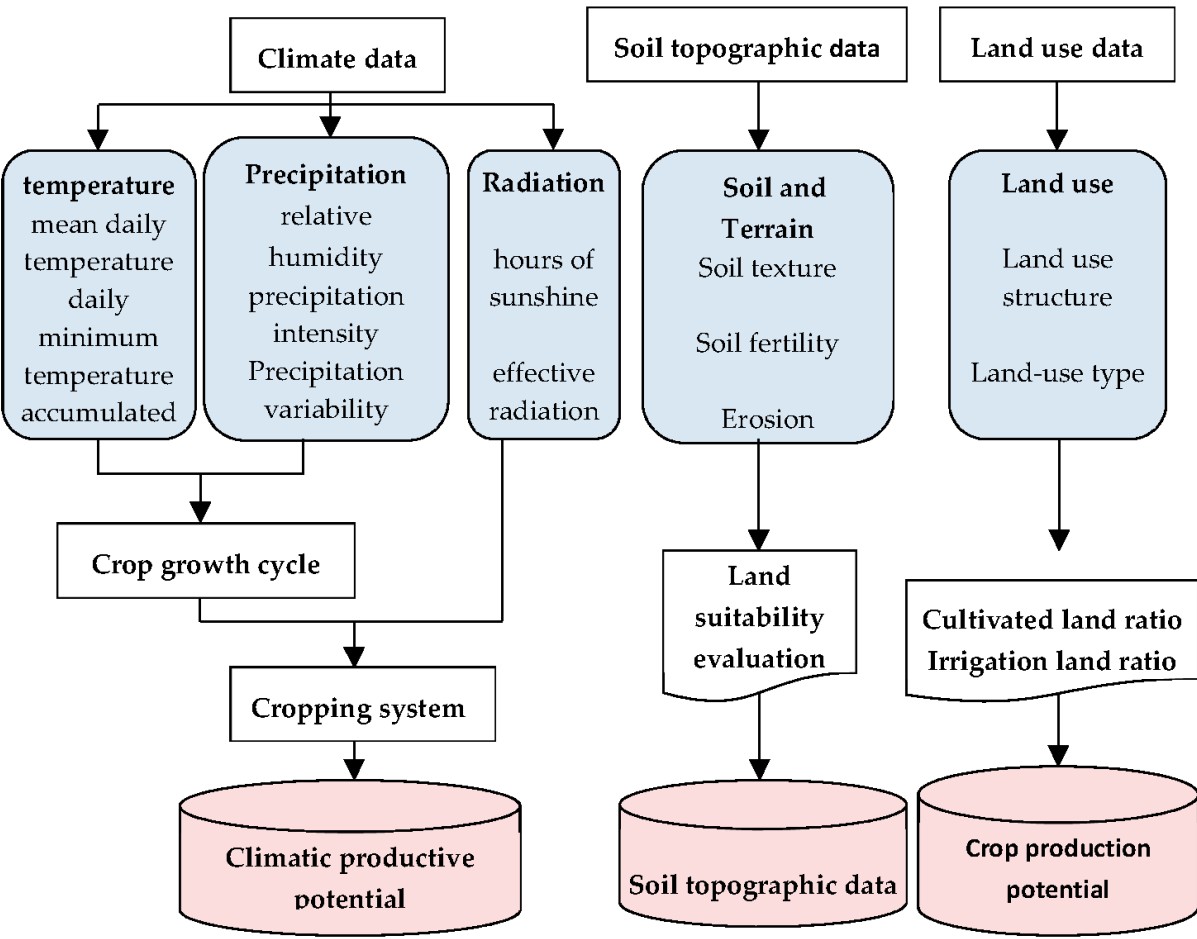

**Figure 2.** GAEZ model structure.

### 2.3.3. Functional Value Evaluation of Ecological Service (*ES*)

In this study, the *ES* function of the land system mainly considers high temperature regulation, air purification, carbon storage, water conservation, tourism and leisure, etc., its total value is:

$$ES = \sum_{i}^{5} ES_i \tag{5}$$

In the formula, *ES* is the value of the *ES* function of the land system, $ES_{1-5}$ is the value of the land system's high temperature regulation, air purification, carbon storage, water conservation, and tourism and leisure functions.

1.  High temperature regulation value ($ES_1$)

The calculation formula is as follows:

$$ES_1 = \sum_{i=1}^{n} \frac{(LST_i - LST_B) \times C \times \rho \times A_i \times H}{EER \times 3.6 \times 10^6} \times P_E \times days \tag{6}$$

Among them, $LST_i$ is the multi-year average summer surface temperature of a certain land use type, $LST_B$ is the multi-year average summer surface temperature of construction land; *C* is the specific heat capacity of the air; $\rho$ is the air density; $A_i$ is the area of a certain land use type; *H* is the average height of residents, This study takes *H* = 1.67(m); *EER* is the energy efficiency ratio of air conditioning and refrigeration, this study takes *EER* =

4.25; $PE$ is the electricity price. In this study, $P_E$ = 0.5 yuan/kwh; days is the multi-year average high temperature days in the study area, that is, the days when the average daily temperature is greater than 25 °C. According to the Xi'an daily average temperature data of China Meteorological Data Network for many years, days = 49.

2.  Air purification value ($ES_2$)

This article mainly measures the purification effect of vegetation on $SO_2$, nitrogen oxides, HF, dust, and other pollutants. The purification amount of air pollutants per unit area of various types of land use is shown in (Table 1). The air purification value is calculated by the following formula:

$$ES_2 = \sum_{ij=1}^{n} A_i \times AP_{ij} \times P_{pj} \tag{7}$$

In the formula, $Ai$ is the area of a certain land use type; $AP_{ij}$ is the purification amount of type $j$ pollutants per unit area $i$ type land use; $P_{Pj}$ is the cost of artificial purification of type $j$ air pollutants per unit weight, according to the "2017 Pollutant Discharge Fee Collection" The standard management method for $SO_2$ and nitrogen oxide treatment costs are respectively 0.6 yuan/kg and 0.63 yuan/kg, fluoride is 0.87 yuan/kg, and other dust is 0.15 yuan/kg.

**Table 1.** Purification rate of air pollutants for each land use per unit area (kg/hm²• A⁻¹).

| Type of Pollution | Forest | Grassland | Water | Unused Land |
|:---:|:---:|:---:|:---:|:---:|
| $SO_2$ | 291.03 | 21.7 | 427.15 | 0 |
| Oxynitride | 215.36 | 16.06 | 316.17 | 0 |
| HF | 9.94 | 1.2 | 3.56 | 0 |
| Other dust | 44300 | 120 | 8.86 | 0 |

3.  Carbon storage value ($ES_3$)

This paper uses carbon storage to characterize the carbon sequestration capacity of the ecosystem (Tallis et al., 2013). This paper uses three major carbon pools; namely, above ground biological carbon ($C_{above}$), underground biological carbon ($C_{below}$), and soil carbon ($C_{soil}$). Specifically, the carbon storage module of the InVEST model uses each land use type as the evaluation unit and multiplies the average carbon density of the three carbon pools by the area of each evaluation unit to evaluate the regional ecosystem carbon storage. The calculation formula is as follows:

$$C_{tot} = C_{above} + C_{below} + C_{soil} \tag{8}$$

In the formula, $Ctot$ is the total carbon density (kg/hm²) of a certain land use type. The value of carbon storage in the land system is:

$$ES_3 = \sum_{i=1}^{n} C_{cot}^{i} \times A_i \times P_c \tag{9}$$

In the formula, $A_i$ is the area of a certain land use type; $Pc$ is the price of carbon sequestration. According to the Swedish carbon tax method in the "Forest Ecosystem Service Function Evaluation Specification" (LY/T1721-2008, this study takes $P_C$ = 1.2 yuan/kg.

Through literature review, the carbon densities of the three carbon pools corresponding to each land type can be obtained, as shown in Table 2 [32].

Table 2. The carbon density of three carbon pools corresponding to each land use (kg/hm²).

| Land-Use Type | Carbon Density of the above-Ground Biomass | Carbon Density of the Underground Biomass | Carbon Density of the Soil |
|---|---|---|---|
| Forest | $4.24 \times 10^4$ | $1.16 \times 10^5$ | $2.37 \times 10^5$ |
| Grassland | $3.53 \times 10^4$ | $0.87 \times 10^5$ | $0.99 \times 10^5$ |
| Water | 0 | 0 | 0 |
| Cultivated land | $0.57 \times 10^4$ | $0.81 \times 10^5$ | $1.08 \times 10^5$ |

4. Water conservation value ($ES_4$)

The value of water conservation in the land system is calculated as follows:

$$ES_4 = \sum_{i=1}^{n} \left( \sum_{j=1}^{3} W_{ij} \times P_w \right) \tag{10}$$

$$W_{i1} = Pre \times l \times A_i \tag{11}$$

$$W_{i2} = f \times q \times A_i \tag{12}$$

$$W_{i3} = h \times k \times A_i \tag{13}$$

Among them, $W_{i1}$, $W_{i2}$, and $W_{i3}$ represent the precipitation interception by the vegetation canopy, the water holding capacity of the litter layer, and the water stored in the soil under the corresponding vegetation type; $Pre$ is the annual average precipitation; $l$ is the interception rate of the forest canopy; $A_i$ is the area of a certain type of land use; $f$ is the dry weight of the litter layer; $q$ is the saturated water absorption rate; $k$ is the noncapillary porosity (Table 3) h is the average thickness of the soil under the $i$-th planting cover type, which is 5m in this study; $P_w$ is the unit storage cost of the reservoir project, which is 6.1107 yuan/m³ [33].

Table 3. Water conservation parameters of each land use.

| Land-Use Type | The Litter is Dry and Heavy (t/hm²) | Saturated Percent Sorption (%) | Canopy Interception (%) | Noncapillary Porosity (%) |
|---|---|---|---|---|
| Forest | 24.56 | 276.45 | 19.35 | 13.46 |
| Grassland | 4.43 | 40.74 | 4.1 | 6.07 |

5. Tourism and leisure value ($ES_5$)

The tourism and leisure value of the land system is revised according to the research results of Xie Gaodi [34] and others:

$$ES_5 = \sum_{i=1}^{n} A_i \times TC_i \times R \times PI \tag{14}$$

$TC_i$ is the national tourism and leisure value per unit area of a certain type of land use (Table 4)

Table 4. Tourism leisure value per unit area of Ecosystem in China (yuan/hm²).

| Land-Use Type | Forest | Grassland | Water | Unused Land |
|---|---|---|---|---|
| Value (yuan/hm²) | 934.13 | 390.72 | 1994.00 | 107.78 |

The $R$ is the biophysical adjustment coefficient, which is the ratio of the average net primary ($NPPs$) productivity of the study area to the national average net primary productivity ($NPPcn$), namely:

$$R = \frac{NPP_s}{NPP_{cn}} \tag{15}$$

Net primary productivity is calculated using the Thornthwaite Memorial model, the formula is as follows：

$$NPP = 3000 \left[1 - e^{-0.0009695(v-20)}\right] \tag{16}$$

$$V = \frac{1.05Pre}{\sqrt{1 + (1 + \frac{1.05Pre}{L})^2}} \tag{17}$$

$$L = 3000 + 25Tmp + 0.05Tmp^3 \tag{18}$$

In the formula, $V$ is the actual annual evapotranspiration; $L$ is the annual average evapotranspiration; $Tmp$ is the multi-year average temperature; $Pre$ is the multi-year average precipitation. $PI$ is a payment index, which is determined by the ability to pay ($AB$) and willingness to pay ($WI$); namely

$$PI = WI \times AB \tag{19}$$

$$WI = 2/(1 + e^{-m}) \tag{20}$$

$$m = \frac{1}{En} - 2.5 \tag{21}$$

$$En = En_r \times (1 - P_u) + En_u \times P_u \tag{22}$$

$$AB = GDP_{ms}/GDP_m \tag{23}$$

In the formula, $En$ is the Engel coefficient of the study area; $En_r$ and $En_u$ are the Engel coefficients of rural and urban areas respectively; $P_u$ is the percentage of urban population; $GDPms$ is the per capita GDP of the study area, and $GDPm$ is the national per capita GDP.

*2.4. Correlation Analysis*

Correlation analysis methods have been widely used in the study of land multi-function trade-offs and synergistic relationships, and used to quantitatively analyze the relationship between various functions and their changing laws over time. In this study, SPSS 22.0 used Pearson's simple correlation coefficient to make a preliminary analysis of the trade-off and synergy between the three types of land function values, and t-test the correlation coefficient. The formula is as follows:

$$\rho_{X,Y} = \frac{cov(X,Y)}{\sigma_X \sigma_Y} = \frac{E((X - \mu_X)(Y - \mu_Y))}{\sigma_X \sigma_Y} = \frac{E(XY) - E(X)E(Y)}{\sqrt{E(X^2) - E^2(X)}\sqrt{E(Y^2) - E^2(Y)}} \tag{24}$$

When the correlation coefficient between the two types of land function values is negative and passes the 0.10 or 0.01 level of significance test, it proves that there is a trade-off relationship between this group of land functions; that is, the increase of a function will leading to the reduction of another function, and when the correlation is positive and the significance test is passed, it proves that there is a synergistic relationship of mutual gain between this group of land functions.

*2.5. Bivariate Local Spatial Autocorrelation Analysis*

Bivariate local spatial autocorrelation analysis is used to measure the spatial correlation of the trade-off and synergy between the two land functions. The bivariate local spatial autocorrelation index (Local Indicators of Spatial Association, LISA) can reflect the

correlation and spatial aggregation degree between the attribute value of a spatial unit and the same attribute value on its adjacent spatial unit:

$$A_i = \frac{1}{n} \frac{(x_i - \bar{x})}{\sum_i (x_i - \bar{x})^2} \sum_j w_{ij} (x_i - \bar{x}) \tag{25}$$

where $x_i$ is the attribute value of geographic unit *i*; $\bar{x}$ is the average of all attribute values; *n* is the total number of geographic units in the study area; $w_{ij}$ is the spatial weight matrix between unit *i* and unit *j*. *LISA* value is used to analyze the concentration of functional value of land system *GP-ED*, *ES-GP*, *ES-ED*. When *LISA* > 0, it shows "high-high" synergy and "low-low" synergy. The former is the area with high relevance of high functional value, and the latter is expressed as the area with high relevance of low functional value; when *LISA* < 0, It is expressed as the trade-off between "high-low" and "low-high", that is, the area where the high function value and the low function value, and the low function value and the high function value have a large correlation; if it is not significant, it shows an independent relationship.

Based on the GeoDa platform, this study uses the Rook proximity principle to determine the spatial weights, and generates a bivariate local autocorrelation map between the three functions of land *ED*, *GP*, and *ES* in 1980, 2000, and 2015. To measure the spatial pattern of trade-offs and synergy between land functions.

### 3. Results

*3.1. Spatio-Temporal Pattern of Land System Functions*

From 1980 to 2015, the *ED* function showed an upward trend. The values were 33.19 million yuan, 42.02 million yuan, and 93.46 million yuan, respectively, and the upward trend was significant from 2000 to 2015, increasing at a rate of 8%; the *GP* function of the land system first declined and then increased, with values of 1124.8 million yuan, 78.51 million yuan, and 10,704 million yuan, respectively, the *ES* function of the land system consisted of high temperature regulation, air purification, carbon storage, water conservation, tourism, and leisure. The sum of the five is declining year by year, which was 239 billion yuan, 237.4 billion yuan, and 236.1 billion yuan, respectively. From 1980 to 2015, the spatial distribution of various land system functions was basically stable, with their own characteristics of change.

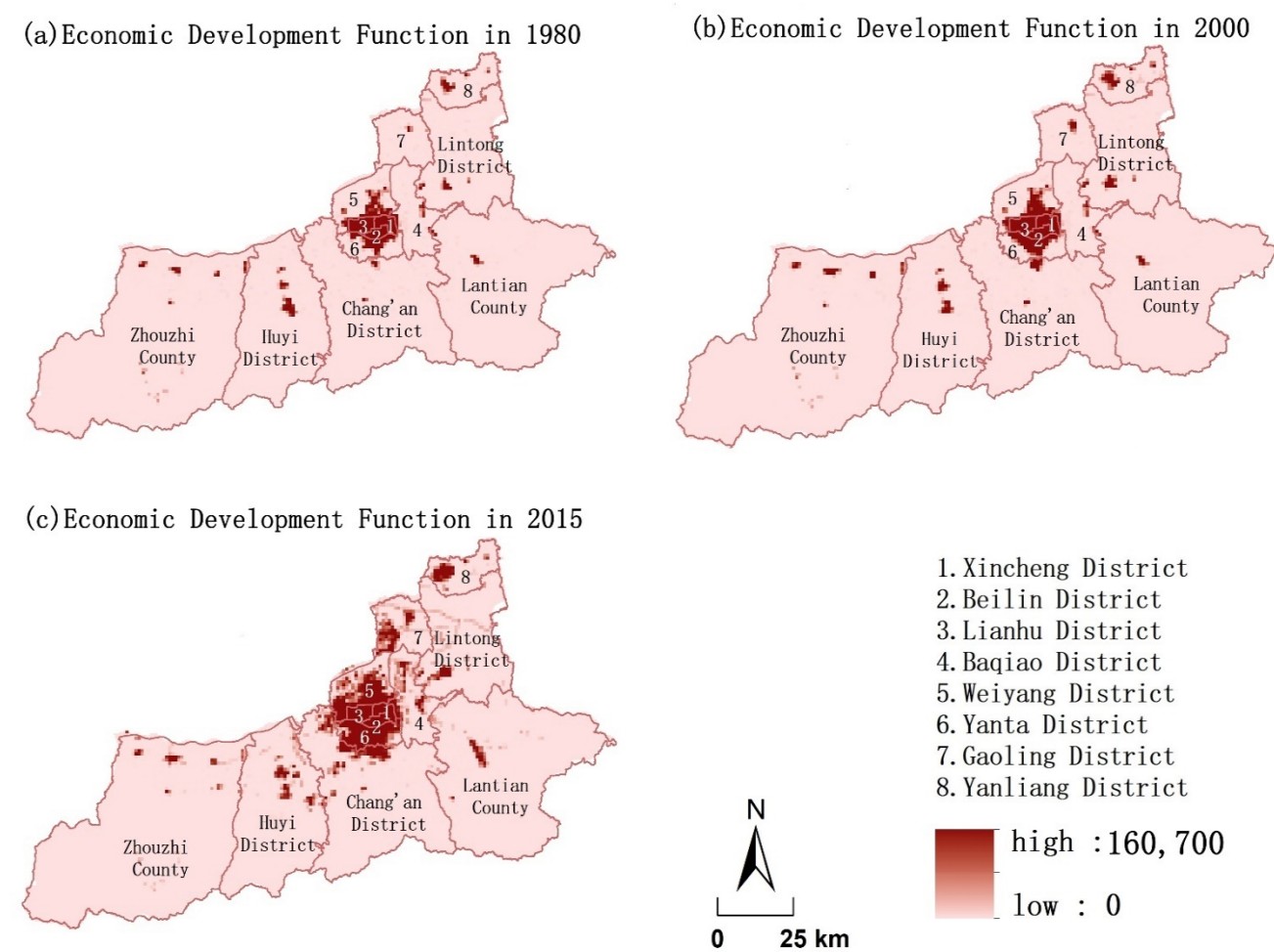

**Figure 3.** Spatial distribution of function of land system economic development in Xi'an from 1980 to 2015 (1 km × 1 km).

The *ED* function of the land system (Figure 3) presented a "low-high-low" pattern from west to east, and its distribution was extremely uneven. The high-value areas were mainly concentrated in the central area, with the highest value being 160,700 yuan. In addition, other high-value areas were mainly concentrated in the central areas of the districts and counties where the population and commercial activities were most dense. The low-value areas were concentrated in the Qinling Mountains, Zhouzhi County, Huyi District, Chang'an District, and the southern side of Lantian County. These areas were dominated by primary industries and had underdeveloped economies. From 1980 to 2000, the land system *ED* function space changed insignificantly, with a slight spread from the downtown area. From 2000 to 2015, urbanization progressed rapidly, construction land expanded rapidly, and the area of high-value areas increased significantly. The central area expanded to the inner and middle areas, and the *ED* function of the land system in the surrounding areas of the Weihe River Basin increased significantly. The *ED* function of the land around the central area of the districts and counties of the city has also been increasing, and the area of high-value areas in the Gaoling District and Yanliang District in the north has also been expanding.

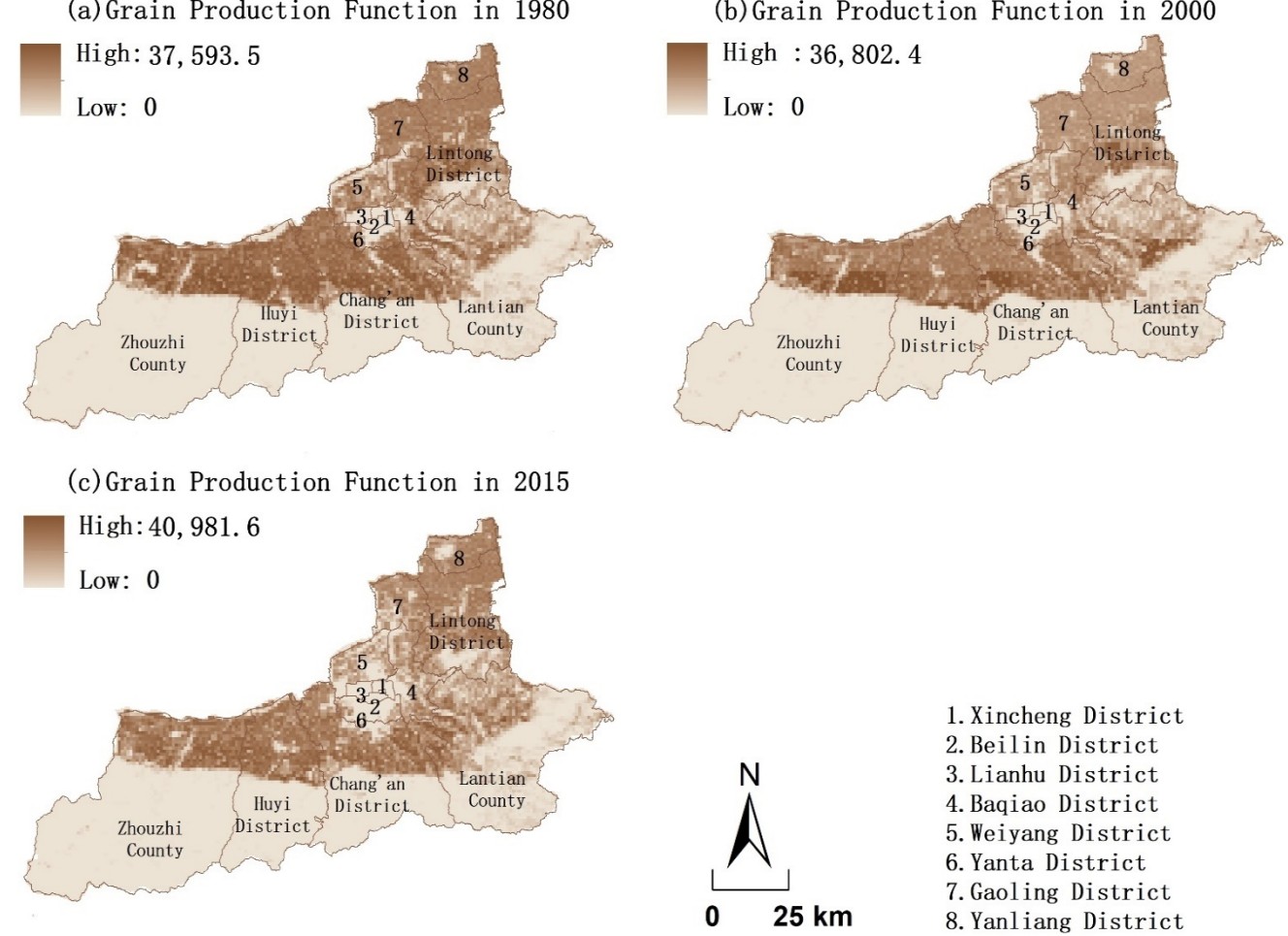

**Figure 4.** Spatial distribution of function of grain production in the land system of Xi'an from 1980 to 2015 (1 km × 1 km).

From 1980 to 2015, the spatial pattern of the *GP* function of Xi'an's land system was basically stable (Figure 4). As the Qinling Mountains lie south of Xi'an with an elevation of 2800 m, Xi'an's general terrain was high in the southeast and low in the northwest and southwest. The cultivated land is mainly distributed to the north of the Qinling Mountains, presenting an incompletely continuous ring pattern of "high-low-high" from west to east. The spatial distribution of high-value areas of the *GP* function of the land system was affected by land use and also presented a pattern similar to the distribution of cultivated land. The highest value was mainly distributed in Zhouzhi County, Huyi District, Chang'an District, and northern Lantian County. The low-value areas were mainly distributed in the central area and the southern area. From 1980 to 2000, the area of cultivated land decreased sharply, and the *GP* function of the land system also decreased correspondingly. Grain issues began to be paid attention to. From 2000 to 2015, with the implementation of rehabilitation and reclamation, the *GP* function showed a gradual upward trend. Its value growth rate was relatively slow (2%).

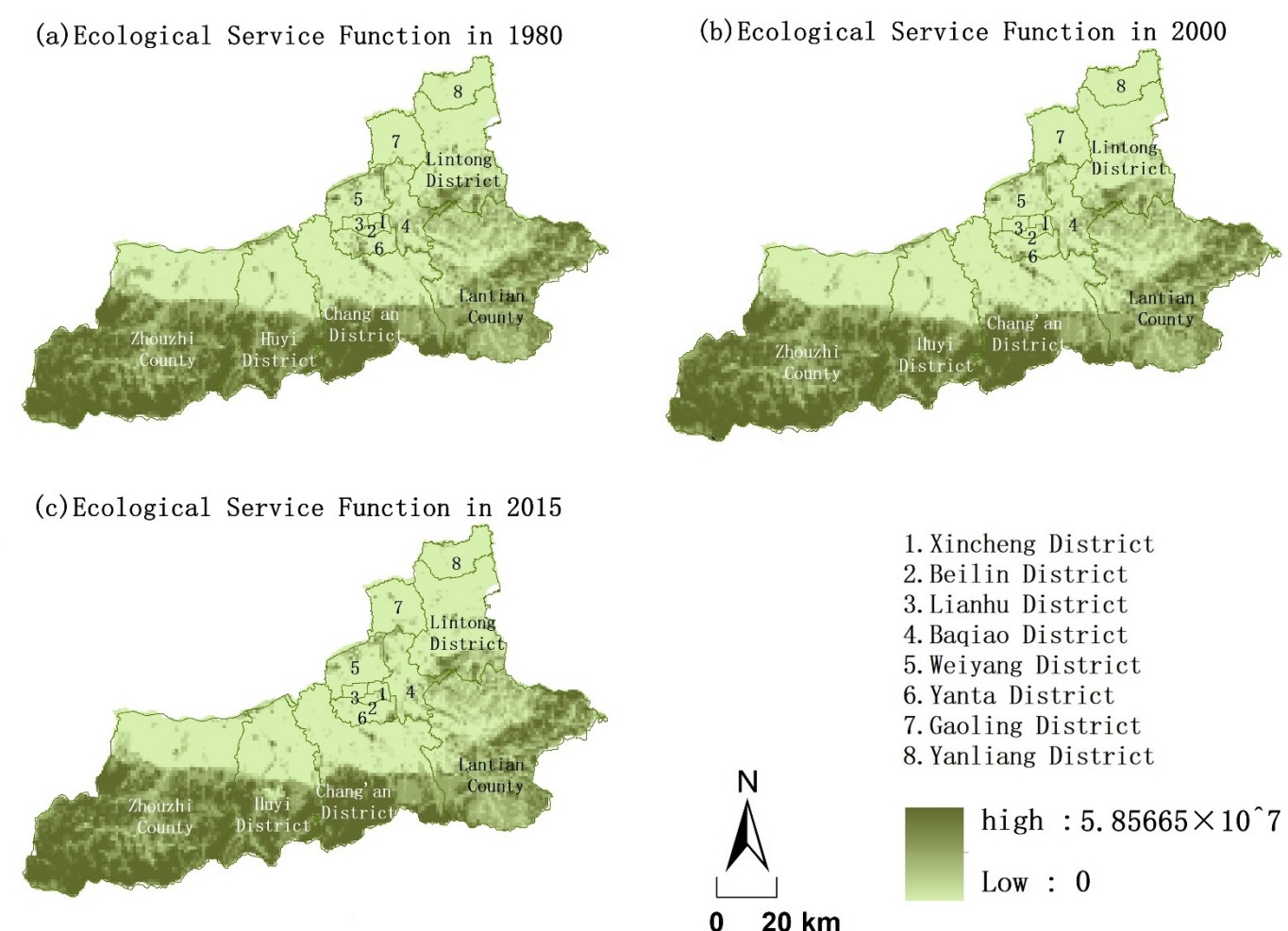

**Figure 5.** Spatial distribution of ecological service value of land system in Xi'an from 1980 to 2015 (1 km × 1 km).

From 1980 to 2015, the *ES* function of the land system in Xi'an showed a declining trend (Figure 5). Among them, the function of high temperature regulation and carbon storage continued to decline, while the function of air purification, water conservation, and tourism and leisure declined from 1980 to 2000, then rebounded slightly from 2000 to 2015 (Table 5). In terms of space, in the Qinling Mountains in the southern part, the land use types were mainly woodland and grassland. Therefore, the high-value areas of land *ES* functions were also concentrated on both sides of the mountains. In addition, Xi'an had a dense network of inland rivers, and *ES* functions were also high. The low-value areas were mainly distributed in the northern part. With the continuous expansion of urban land, the ecological space was seriously threatened, and behaviors that destroyed the ecological environment, such as illegally built villas in the Qinling Mountains, had repeatedly broken the limit of environmental carrying capacity. In addition, a large number of woodland and grassland reclaimed to supplement cultivated land. As a result, the *ES* function of the land system showed a gradual decline.

**Table 5.** Functional value of Land system Ecological Service in Xi'an from 1980 to 2015 (yuan).

| *ES* Function Value (yuan) | High Temperature Adjustment | Air Purification | Carbon Storage | Water Conservation | Travel and Leisure |
|---|---|---|---|---|---|
| 1980 | 4665 | 212,971 | 20,199,020 | 3,423,101 | 64,687 |
| 2000 | 4564 | 211,744 | 20,052,318 | 3,403,714 | 63,076 |
| 2015 | 4539 | 214,483 | 19,904,660 | 3,425,488 | 63,748 |

*3.2. The Trade-Off and Synergy between the Functions of the Land System*

3.2.1. Correlation Coefficient Analysis

The study used Person's simple correlation coefficient to analyze the functions of three types of land systems in Xi'an. From 1980 to 2015, the correlation coefficients between the values of the three functions all passed the 0.01 significance level test. It could be seen from Table 6 that the correlation coefficients among the land system *ES*s-*GP* function, *ED*-*ES* function, and *GP*-*ED* function were all less than 0, and it also showed that the three land system functions were in a trade-off relationship; that is, the increase of one function will lead to the decrease of the other two functions. Among them, the absolute value of the correlation coefficient between *ES*s and *GP* functions was close to 1; that is, the negative correlation between the two was strong, the trade-off effect was more significant, and the conflicts and contradictions reflected in the space were also more intense. The absolute value of the correlation coefficient between *GP* and *ED* function was closer to 0; that is, the correlation between the two was weak, and its trade-off effect was not significant.

In terms of time, from 1980 to 2015, the absolute value of the correlation coefficient between *ED*-*ES* function and *ED*-*GP* function gradually increased; that is, the trade-off between the two functions gradually increased, the absolute value of the correlation coefficient between *ES* and *GP* functions was gradually decreasing; that is, the trade-off between the two functions was gradually weakening; that is, the spatial conflicts and contradictions between the two functions were alleviated.

**Table 6.** Correlation coefficient between functional values of land system in Xi'an from 1980 to 2015.

|  | *ES* | | | *GP* | | |
|---|---|---|---|---|---|---|
|  | **1980** | **2000** | **2015** | **1980** | **2000** | **2015** |
| *ED* | −0.156 ** | −0.177 ** | −0.275 ** | −0.028 ** | −0.048 ** | −0.063 ** |
| *ES* | | | | −0.741 ** | −0.714 ** | −0.697 ** |

** Significantly correlated at 0.01 level (two-sided).

3.2.2. Space Trade-Offs and Synergy

This study used GeoDa to perform bivariate local autocorrelation analysis of three land system functions, and further drew LISA diagrams to obtain the spatial pattern of trade-offs and synergy between land system *ES*-*GP* functions, *ED*-*ES* functions, and grain production-*ED* functions from 1980 to 2015. Among them, the high-high and low-low matching regions were synergistic regions for mutual gain, and the high-low and low-high matching regions were trade-off regions.

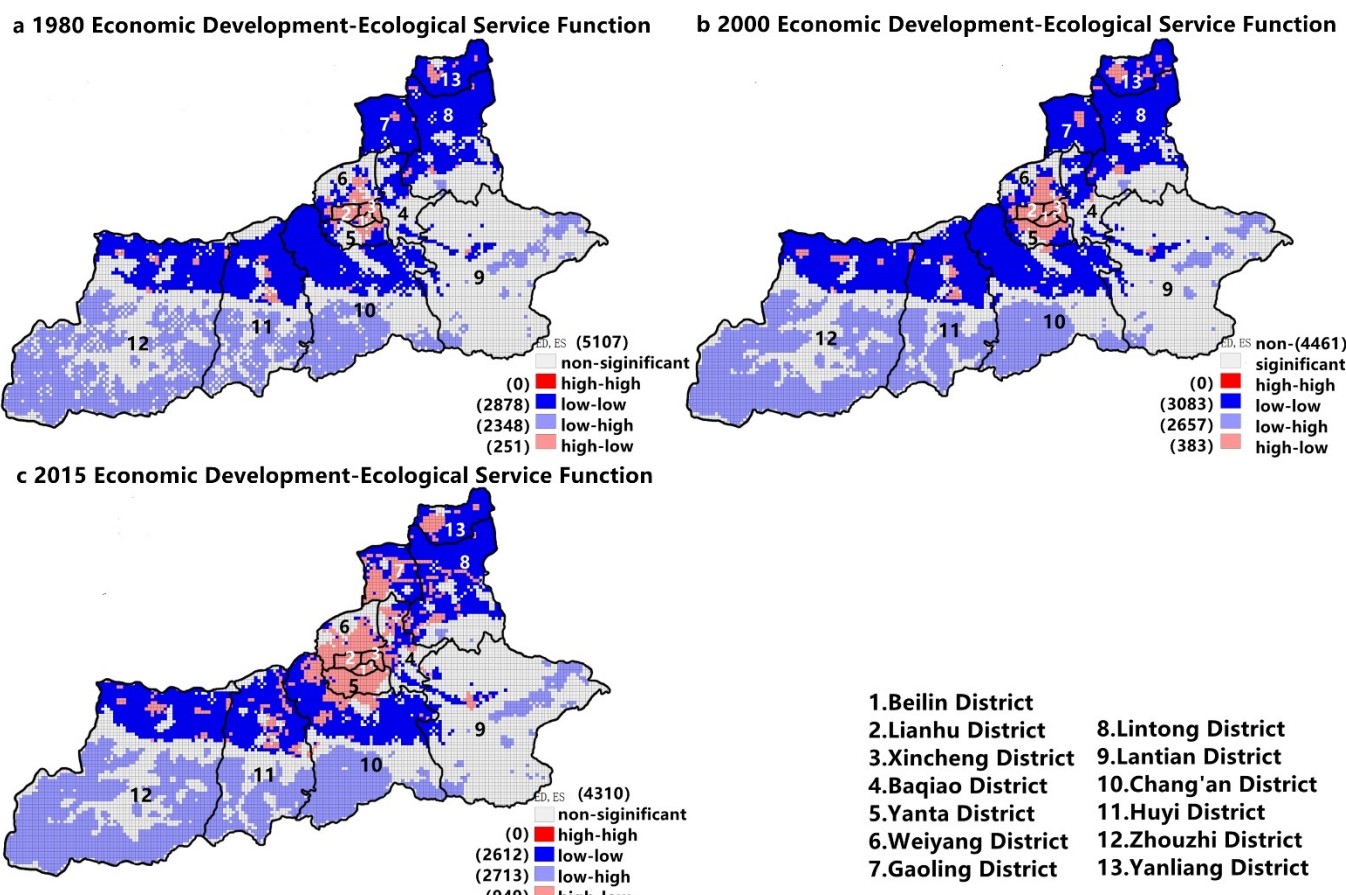

**Figure 6.** LISA Map of Land System Economic Development—Ecological Service Function. In Xi'an city from 1980 to 2015.

It can be seen from Figure 6 that the area of the synergy zone and the trade-off zone for the *ED* of the land system in Xi'an and the *ES* function were roughly the same. They were 27.19%, 29.13%, and 24.46%. In 1980 and 2000, the area of the synergy area was slightly larger than the trade-off area, and in 2015, the area of the trade-off area was larger than the synergy area. The synergy was mainly reflected in the synergy between low and low values, and there was no "high-high" synergy. In terms of spatial distribution, the "low-low" synergies of *ED* and *ES* functions were distributed in a ring around the center of Xi'an. Among them, the "low-low" synergy of Yanliang District, Gaoling District and Lintong District was significant, these areas were mainly cultivated land, construction land was scattered among them, and their land system *ED* functions and *ES* functions were low, so they presented a "low-low" synergy pattern. The "high-low" trade-off of *ED* and *ES* functions of Xi'an's land system were concentrated in Weiyang District, Beilin District, Yanta District, Lianhu District, and other areas. It was the most concentrated area of population and commercial activities. High-end retail districts and commercial streets were densely distributed, construction land was concentrated, but green area was insufficient, and ecological land such as woodland and grassland was less. Therefore, the *ED* function of the land system in this area was high, and the *ES* function was low, so it presented a high-low trade-off pattern. In the southern part of Xi'an and the Qinling Mountains, the land use types were mainly woodland and grassland. The *ES* function was high, and the *ED* function was low. Therefore, the region presented a low-high synergy. From 1980 to 2015, overall, the area of the trade-off between *ED* and *ES* functions of Xi'an's land system increased significantly, spreading from the central area to surrounding districts and counties, indicating that the *ED* function of Xi'an's land had increased, while the *ES* function had fallen.

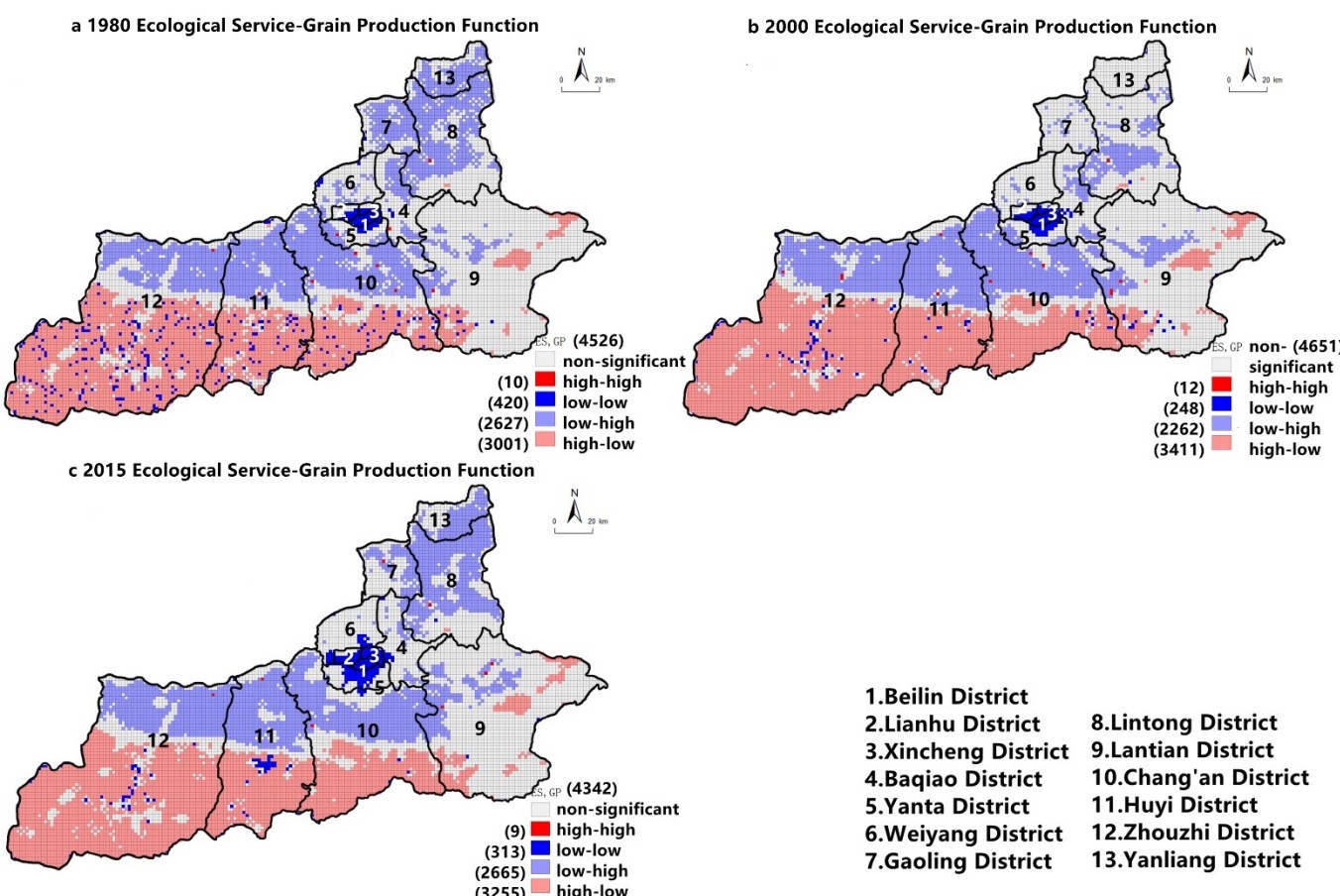

**Figure 7.** LISA Map of land system Ecological Service—Grain Production Function in Xi'an city from 1980 to 2015.

*ES-GP* function in Xi'an was dominated by the trade-off area. From 1980 to 2015, the area of the trade-off area accounted for 53.17%, 53.60%, and 55.93%, and the area of synergy areas accounted for 4.06%, 2.46%, and 3.04%. The trade-off between the two was significant. The low-high and high-low trade-off areas were concentrated in the Weihe alluvial plain north of the Qinling Mountains and the Qinling Mountains (Figure 7). The Qinling Mountains had less cultivated land and large woodlands and grasslands, so the *ES* function was higher, the *GP* function was low. The land use types north of the Qinling Mountains were mainly cultivated land and construction land, with high *GP* function and low *ES* function. Due to the implementation of the policy of returning farmland to forests, many scattered low-low synergistic areas in the Qinling Mountains gradually transformed into high-low trade-off areas; that is, *ES* functions have increased.

The synergy and trade-off between the *GP* and *ED* functions of the land system were not significant. As shown in Figure 8 the areas with insignificant correlation between the two accounted for a relatively large area, but overall the area of the trade-off area was larger than the synergy area; among them, the low-high trade-off area was concentrated in the center, the most densely populated area, and the high-high synergy area was concentrated in the center and the fringe areas of the centers of districts and counties, mostly large villages, the "high-low" trade-off areas were scattered in Lintong District, Lantian County and other places. The land use types in these areas were mainly cultivated land and there were few commercial activities. From 1980 to 2015, the area of the trade-off between the two showed an increasing trend, and the increase was more obvious from 2000 to 2015.

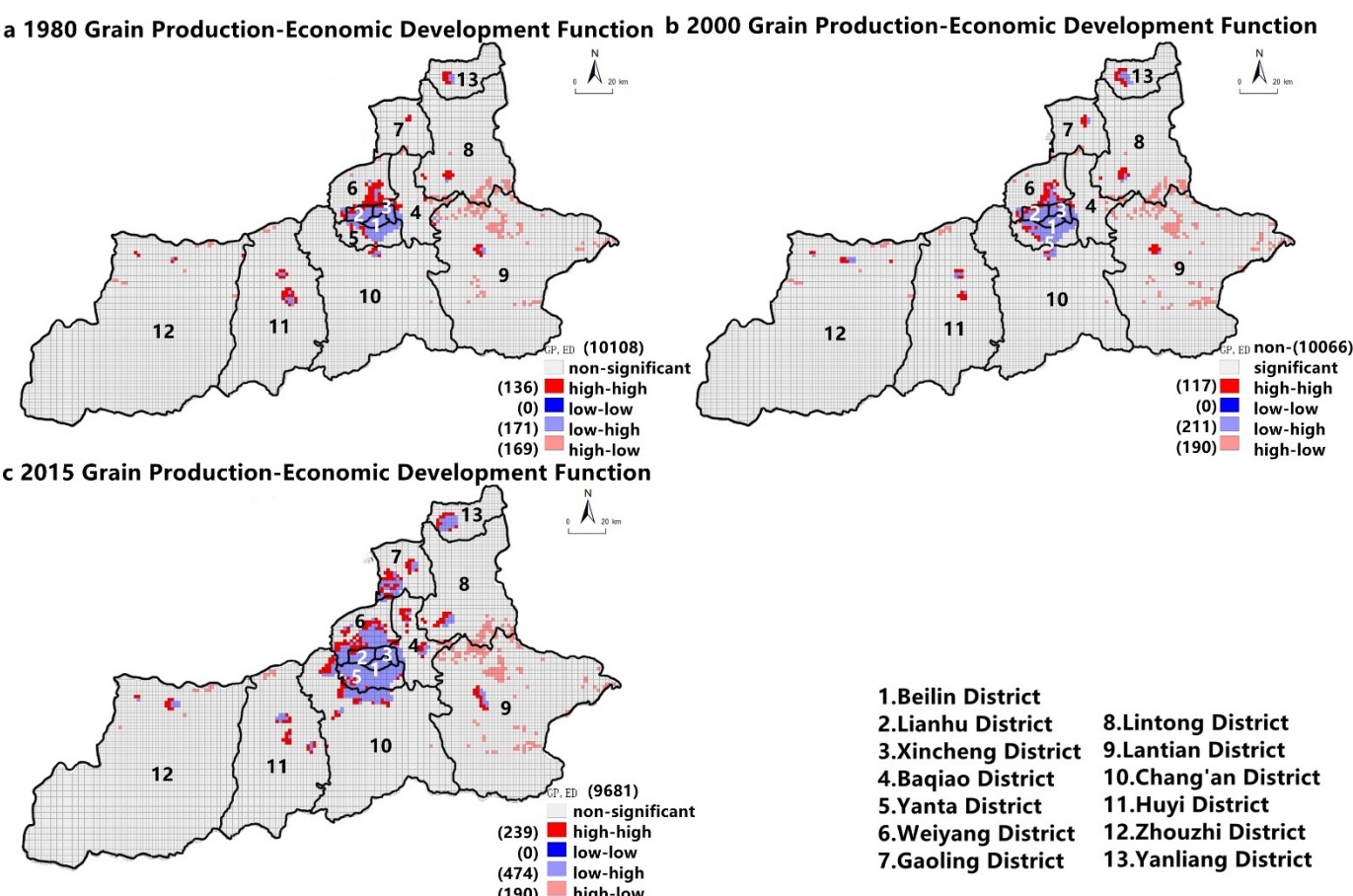

**Figure 8.** LISA map of the Land system Grain Production-Economic Development Function of Xi'an city from 1980 to 2015.

## 4. Discussion

### 4.1. The Impact of Urban Expansion on the Temporal and Spatial Patterns of Functions

Urban expansion is the most significant feature of current global land use changes [35] During 1980–2000, the continuous expansion of urban development space brought about a significant increase in the *ED* function of land system. Its high-value areas spread from the city center to the surrounding areas, mainly caused by the rapid progress of urbanization and the expansion of construction land. During this period, the city was expanding blindly and the layout is chaotic. construction land continued to squeeze cultivated land, leading to a sharp decline in high-value areas of *GP* value. In addition, the rare extreme drought climate in 1997 also caused a nationwide reduction in *GP*, grain security is seriously threatened. The current proportion of cultivated land in my country is only 12.76%, which is much lower than the United States, India, and other countries, and due to the expansion of cities, this ratio is decreasing year by year; the cultivated land occupied by construction land was mainly distributed in the eastern region, and the quality was good. The increased cultivated land was mainly marginal land of poor quality, and the quantity and quality were significantly decreasing, which directly affected a country's food security and social stability. [36] So ecological land such as forest land has been used to supplement cultivated land, resulting in a significant decline in *ES* functions. In addition, urban expansion was also seriously encroaching on land use types with high ecosystem service values such as woodlands, grasslands, and wetlands, causing serious damage to some of the original functions of these ecosystems, such as soil conservation [37], climate regulation [38], etc. Combining with the land use transfer matrix made by Ma Xinping et al. [39], it can be seen that in this period, the land use types were mainly converted from forest land to cultivated land and cultivated land to construction land, the proportion of transfers accounted for 74% and 71.99%, respectively. It also confirms the

conclusion of this research. In this stage, construction land cannibalizes cultivated land, and ecological land continues to replenish cultivated land, causing intense conflicts in the "production, ecology, and living" space[40], at the expense of *ES* functions, the continuous *ED* and urban expansion have seriously endangered the sustainable development of the land. During 2000–2015, with the urbanization process has stabilized, food security issues became increasingly serious, and re-cultivation was implemented. Also, the government continuously improved the land law enforcement, supervision system, and mechanism, and increased the financial input of the main grain producing areas, so the function of *GP* increased. Since then, the increase in *GP* functions would depend more on the increase in yield, rather than just on the increase in area, to ease the situation of cultivated land tension. Countries all over the world are committed to protecting cultivated land, especially the United Kingdom, Japan, and other countries where the man–land relationship is more tense than in China. Its policies and measures are worthy of our reference and consideration. The United Kingdom attaches great importance to the protection of agricultural land. As early as after World War II, it began to encourage the development of farms on a large scale and gave farmers certain subsidies. Since then, it has paid more attention to improving the quality of the rural environment and developing the rural economy, as well as restricting urban expansion. Therefore, the protection of agricultural land in the UK is mainly through a flexible planning system, rather than just focusing on the protection of farmland [41]. The basic national conditions of the "small peasant economy" under China's specific historical conditions have determined that my country's agriculture must follow a gradual, moderate-scale operation, and it must not be rushed [42], and pay attention to protecting the economic vitality and environmental value of the village. Japan's metropolitan structure reduces the waste of land due to repeated construction and is also a way to ease the tension on cultivated land. During this period, the *ES* function of the land system was still in a downward trend. Combined with the analysis of the ecological environment adaptability of the northern foot of the Qinling Mountains (Xi'an section) by Xiao Zhetao et al. [43], it can be seen that the northern foot of the Qinling Mountains, as an ecological ecotone, has extremely important ecological significance. However, due to the pursuit of commercial interests, a large number of real estate projects spread across the Qinling Mountains, constantly breaking through and eroding the ecological environment of mountains and waters, disturbing the habitat of animals, and breaking the ecological balance. According to statistics, the value of global *ES* functions is more than 33 trillion dollars each year, but the rapid and unreasonable urbanization process has strongly disturbed the ecosystem and caused the degradation of natural ecosystem service functions [44]. The degradation of urban ecosystem service system functions is generally reflected in the fragmentation of the landscape and the obvious changes in the species structure on a small scale. The urban ecosystem is gradually losing biodiversity [45], causing the decline of the standard of living and welfare of urban residents. The degradation of ecological functions caused by urban expansion has become a global problem.

### 4.2. The Impact of Urban Expansion on Functional Trade-Offs

In terms of time, the balance between *ES-ED* functions and *GP-ED* functions of the land system from 1980 to 2015 has been increasing. This is one of the important issues that currently exists. According to the logic of the Environmental Kuznets Curve (EKC) [46], per capita income and the degree of environmental degradation present an inverted U-shaped relationship; that is, with the continuous development of the economy, the degree of ecological environmental degradation first rises and then declines. Developing countries need to go through the development path of *pollution first*, *governance later*, so ecological deterioration at the beginning of economic growth is inevitable; the development of *GP* and agricultural economy is of great significance to the economic growth of the entire society. From a historical point of view, the industrial revolution in western developed countries developed on the solid foundation of the agricultural revolution. The develop-

ment of industry in turn promoted the progress of agriculture to increase agricultural labor productivity, and the output of major agricultural products such as grain increased substantially [47]; however, with the further development of industrialization and urbanization, the synergy between *GP* and *ED* has become weaker. Instead, the *ED* has been blindly pursued at the cost of agricultural shrinkage and the abnormal development of the national economy.

In terms of spatial distribution, the *ED-ES* functions of the trade-off area and the synergy area are relatively close. The trade-off areas are concentrated in the center of Xi'an and the Qinling Mountains in the south. They are reflected in the pattern of high economy-low ecology and low economy-high ecology, the synergy zone is distributed in a ring around the city center. The area of the trade-off area between the two tends to increase year by year, and the spatial conflict between the two is mainly intensified in the central area of the districts and counties and the Qinling Mountains. Research by Liu Pengfei and others also pointed out that [48], the large contrast between the quality of production space and ecological space is a common problem in most cities in China at this stage, especially the southeast coastal cities. Although the quality of production space is high, the quality of living space needs to be improved.

The *ES*-food production function trade-off area is significantly larger than the coordination area, is the most significant problem of land use in the study area, and the spatial contradiction between the two is relatively large. The significant conflict between these two functions mainly comes from the spatial contradiction between the two types of land use, cultivated land, and forestland. To coordinate the spatial relationship between these two types of land, we must first measure their respective values. The value of cultivated land mainly includes the following aspects [49]: firstly, the income of the food crops it grows; secondly, due to the large population of our country, cultivated land also bears the social responsibility of ensuring food security and arranging rural labor; as an environmental landscape, cultivated land also has environmental value. For forest land, its value first includes a certain economic value; that is, it can provide a variety of forest products, but the non-market value of forest land is far greater than its economic benefits. As a natural ecosystem, it can conserve water, regulate climate, prevent wind and fix sand, purify the air, and is also a habitat and breeding place for various wild animals and plants. It plays a huge role in protecting the environment and maintaining ecological balance. In addition, its cultural landscape value and aesthetic value cannot be ignored. To summarize, both types of land have their own unique value. Blindly seeking ecological benefits and returning farmland to forests will inevitably affect food security; while blindly pursuing food production and occupying ecological land will inevitably destroy the ecological barrier. Therefore, it is necessary to coordinate various interest relationships, choose steep slopes with a lower input-output ratio as much as possible when returning farmland to forests, and increase the productivity of remaining farmland so that food security, farmers' economic interests, and ecological environmental benefits are balanced.

### 4.3. Implications, Applications, and Limitations

There was a significant trade-off between *GP* and *ES* functions of the land system, which is an important issue facing Xi'an land use. In addition to the above-mentioned, due to the continuous expansion of construction land, the occupation of cultivated land, and the repeated breakthrough of the grain red line, a large amount of ecological land reclaimed to supplement the cultivated land; in addition, the changes in the ecological environment will also cause the quality of the soil to decline, which in turn will affect the grain yield, causes the conflict between the *GP* function and the *ES* function of the land system. Zhou Zhongxue [50] believes that the fragmentation of landscapes such as cultivated land, garden land, and woodland in rapidly urbanized areas is the cause of the substantial decrease in *ES* in the Xi'an metropolitan area, which also complements the conclusions of our study and reveals the impact of farmland fragmentation on *ES*. Therefore, how to achieve the coordinated development of the two functions is a key issue that needs

urgent attention at present. In order to solve the spatial contradiction between cultivated land and ecological land, under the premise of limited cultivated land area, improving the cultivation rate and efficiency of cultivated land has become the key to alleviating the spatial conflict. While strictly protecting cultivated land and the number of basic farmland, more should develop high-quality basic farmland; at the same time, intensive use of agriculture should be vigorously promoted to promote the high-quality development of agriculture, the government needs to effectively strengthen the construction of agricultural infrastructure, and provide farmers with appropriate subsidies and technical training, paying attention to the directivity and accuracy of subsidies [51], actively move closer to green and ecological agriculture, and aim at ecological protection and food security, so as to increase *GP* and farmers' income simultaneously, which is more conducive to the enthusiasm of farmers and increase the cultivation rate. Zhang Liqing [52] put forward that facing the tense human-land relationship and severe food security situation in Xi'an, the use of intensive and ecologically balanced cultivated land is an inevitable choice for sustainable development. That is, strengthening the ecological protection of cultivated land, developing a circular economy, and reducing energy consumption are feasible strategies to solve the spatial contradiction between cultivated land and ecological land. The non-agriculturalization of cultivated land is also one of the problems to be solved in the protection of cultivated land, which must be strictly controlled and implemented in the village. In addition, protecting and improving the ecological land is imminent. It was necessary to ensure ecological safety barriers, especially the ecological management of the northern foot of the Qinling Mountains, such as the establishment of concentrated and contiguous nature reserves. It is also necessary to increase the greening rate of the city and increase the area of forest land. In addition, the overall quality of food producers is generally not high, which is also one of the causes of environmental degradation and other problems. Therefore, it is necessary to rely on technological innovation to make breakthroughs, and to rely on professionals to improve. Only in this way can a virtuous circle of prosperous food and good ecology be formed. In recent years, all localities are actively responding to the call of *Green water and green mountains are golden and silver mountains*, and in the division of the *three areas and three lines* of the national land and space plan, the delineation of the ecological protection red line is the first place; under the adjustment and implementation of policies such as strict protection of basic farmland and promotion of high-quality agricultural development, the trade-off relationship between *GP* and *ES* functions of land system was gradually weakened, and the contradiction between the two had a tendency to ease, proving that the treatment of food and ecological problems has begun to show results.

The trade-off between *ED* and *ES* functions was weak. The *ED* function was mainly related to the expansion of construction land, while the *ES* function was closely related to ecological land. The spatial pattern of these two types of land was relatively stable, construction land was mainly distributed in the middle area of the Weihe Plain at the northern foot of the Qinling Mountains, while woodland Grasslands were mainly distributed in the Qinling Mountains in the south, and there was less conflict between the two. However, in recent years, with the continuous intensification of urbanization and urban land expansion, the resulting air pollution, heat island effect, climate change, and other phenomena had also profoundly affected the ecological environment, resulting in a certain degree of degradation of *ES* functions. In addition to the influencing factors mentioned above, Qin Yanli et al. also mentioned that the expansion and spread of urban architectural landscapes, and the construction and extension of main traffic arteries have aggravated the fragmentation of the landscape, leading to a reduction in soil conservation and climate regulation functions that account for a high proportion of ecosystem services; in addition, a large amount of sewage discharge in industrial development has led to a decline in the quality of the soil, resulting in a reduction in vegetation coverage, which has also led to a continuous decline in *ES* functions. This requires that while accelerating urbanization, attention is paid to its negative effects on the ecological environment, such as

controlling the pollution problems of related enterprises and factories, managing water sources, and improving soil fertility to cope with grassland degradation. At the same time, it is also necessary to increase the diversity of woodland patches to improve the ecological environment.

There was also a weak trade-off relationship between *GP* and *ED* functions. The function of *GP* was closely related to cultivated land, and cultivated land and woodland were staggered in the northern part. Therefore, as the city expanded, a large amount of cultivated land was occupied by construction land, rural labor force was lost, and a large amount of them flowed from primary industry to tertiary industry, resulting in abandonment of cultivated land. The phenomenon was serious, leading to a decline in *GP* functions. In addition, the continuous growth of population has led to the need for more arable land to meet rations, and at the same time, more arable land is needed to build infrastructure to meet the needs of production and life, resulting in a vicious circle of contradictions between arable land and construction land. This is consistent with the conclusion drawn by Jia Shunan [53] and others; that is, the *ED* of Xi'an City has caused damage to the ecological environment of cultivated land to a certain extent. Solving the problem of non-agriculturalization of arable land and reducing the loss of a large number of rural laborers to cities is the key to solving the contradiction between food production and *ED*. It is necessary to increase the control of non-agricultural construction land occupation of cultivated land and improve reporting channels; and to guide and train farmers to adopt advanced technology, adopt the method of *replacement of subsidies with rewards* to stimulate farmers' enthusiasm. For countries such as South Korea, Japan, and other countries where the supply and demand of land resources are tight, scholars mainly focus on the study of the factors affecting the non-agriculturalization of cultivated land in the aspect of cultivated land [54]. In terms of construction land, it mainly studies the phenomenon and problems of land speculation and skyrocketing land prices [55], their purpose is to highlight the importance of intensive land use. However, urbanization construction has different demands for different types of land. It would be biased to study the relationship between the two types of land only for a single type of land (cultivated land or construction land) [56]. The issue of cultivated land and construction land in China should start from the coordination of the two.

### 4.4. Explanation of the FOUR Research Gaps

In response to the current research gaps mentioned in the introduction, this research has made the following supplements:

At present, the construction of the system of land multi-function trade-off research is not yet mature, so this study did not choose the method of establishing the index system that was used by researchers in the past. Instead, we directly established a calculation model for the three functions of the land system. Based on the vector data of each period of land use, combined with climate, soil, and socio-economic data, the value of each function of the land is evaluated, and abstract functional concepts are directly converted into specific values, revealing the true change of the land. In addition, when the index system is used for research, it is mainly through some socioeconomic statistical data to perform mathematical operations, of which it is difficult to show the temporal and spatial change pattern of the trade-off relationship of functions. The selection of indicators is also affected by subjective factors to a certain extent, and indicators selected by different researchers are quite different, making it difficult to make a horizontal comparison between various studies.

Normal research poses human needs as the main body of research, such as selecting the indicator of economic function as *per capita output value of secondary and tertiary industries*, etc., and paying less attention to the land itself. The classification of land functions used in this study is based on the attributes of the land itself; that is, the functions and services it can provide for human beings. The *ED* function takes into account the impact of the expansion of construction land on *ED*; the *GP* function is also based on the potential

of the land itself to provide food; and the *ES* function takes into account the forest land, grassland, and water areas with high ecological value in the land. Analyzing the services that various land types can provide can reflect the real situation of the land.

At present, there are abundant researches on the balance of ecosystem services in the field of ecology. This research is to apply the correlation analysis commonly used in the weighing methods of ecosystem services to the weighing of the various functions of the land system, and it can also reflect the spatial pattern of the trade-off relationship between the various functions, but its internal mechanism still needs to be further explored.

However, this study only selected the three phases of 1980, 2000, and 2015 data for analysis from 1980–2015, which cannot fully reflect the multi-functional temporal and spatial evolution of the land system in this time series, and has certain limitations. In addition, the correlation coefficient method and bivariate local autocorrelation analysis used in this article can only initially analyze the spatial distribution of trade-offs and synergy, but cannot reveal the internal mechanism of trade-offs and synergy. In the future, we will combine multiple periods of land use data to deeply explore the mechanism behind the multi-functional trade-off and synergy of the land system, and further draw targeted decision-making recommendations.

## 5. Conclusions

Research on multi-function trade-offs of the land system could not only enrich the research system of land system change and deepen the understanding of the multi-function of the land system, but also help to understand the relationship between various functions of the regional land system. Reasonable trade-offs were made in land management and utilization decision-making to realize the sustainable use of land resources and the sustainable development of social economy.

This study took Xi'an as a case study, and by quantitatively evaluating the value of the *ED*, *GP*, and *ES* function of the land system, this paper discussed the time dynamics and spatial pattern of the multi-functional space trade-off caused by the changes in the land system in Xi'an since the reform and opening up. In this study, we developed a general model to assess the value of the *ED*, *GP*, and *ES* functions of land system, and learned from the research methods of ecosystem function trade-offs, applying bivariate local autocorrelation analysis to explore land function trade-offs. Based on this, we got the spatial pattern of the three functional values and their trade-offs over time.

On this basis, we concluded that in the context of rapid urbanization, the economic development function of land system continues to increase, accompanied by the first key problem of the continuous decline of ecosystem services. The second key issue worthy of attention is the significant trade-off between grain production and ecological services. The trade-off between economic development-grain production function and economic development-ecological service function is also one of the common problems in the rapid urbanization area.

Our data suggested that we still have a long way to go. The decline of ecological service function of land system in rapid urbanization area indicated that it is urgent to protect and improve ecological land. All regions need to improve the greening rate of the city and fundamentally increase the *ES* function; in order to reduce the negative effect on ecological land, strict laws should be introduced to limit the emission of factories; in urban design and planning, landscape fragmentation caused by disorderly spread should be avoided; and the establishment of concentrated and contiguous nature reserves, etc. The significant trade-offs of *GP-ES* function was a key issue for countries with tight supply and demand of land resources, because it involved not only the most basic human food security issues, but also the human settlement environment. Therefore, governments need to coordinate various interest relations, choosing steep slope land with low input-output ratio to return farmland to forest, so as to increase *ES* function while ensuring grain yield. In order to balance the function of *GP-ES*, we should improve the productivity of surplus cultivated land by means of scientific and technological innovation, subsidize farmers,

improve the comprehensive quality of grain producers, and moderate scale operation. The trade-off between *ED-ES* and *ED-GP* was increasing year by year, which was also one of the facts worthy of vigilance in the areas of rapid urbanization. In some countries where the conflict between people and land is significant, we should not only deal with a single type of land, but should start with the coordination of two kinds of land (construction land and cultivated land, construction land and forestland).

The relationship between the five functions of the *ES* needs to be further explored and it needs data with a smaller time span and more periods to support this conclusion. Nowadays, we still use the method of multi-function trade-off of ecosystem to study the land system. In the future, we need to find more scientific and reasonable methods to study the trade-off relationship between multi-functions of the land system in order to reveal its internal mechanism. Moreover, all countries can make a partition study on the trade-off relationship between different types of land functions, which can better adapt to the national conditions and put forward more directional policy recommendations. This is desirable for future work.

**Author Contributions:** Conceptualization, J.S. and F.L.; methodology, F.L.; software, J.S.; validation, J.S. and F.L.; formal analysis, J.S.; investigation, J.S.; resources, J.S.; data curation, J.S.; writing—original draft preparation, J.S.; writing—review and editing, J.S.; visualization, J.S.; supervision, J.S.; project administration, J.S.; funding acquisition, F.L. All authors have read and agreed to the published version of the manuscript.

**Funding:** This work is supported by National Natural Science Foundation of China (NSFC) (Grant number. 41701094).

**Data Availability Statement:** The data presented in this study are openly available in [China Meteorological Data Network, http://data.cma.cn/site/index.html (accessed on 2020/05/30)].

**Conflicts of Interest:** The authors declare no conflict of interest.

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
