# Peer review of "Multi-Function Tradeoffs of Land System in Urbanized Areas—A Case Study of Xi’an, China"

_land, doi:10.3390/land10060640_

Round 1

Reviewer 1 Report

Major comments:

From the point of view of the carried analysis, it is right to use the 1 x 1 km grid, and not to refer to the boundaries of territorial units. However, the choice of the shape and size of the assessment field (1 km square) was not justified. Why did the Authors decide to choose squares, and not, for example, hexagons, which are a better spatial evaluation field?

In my opinion, the selection of literature is also biased, most of the references are published by Authors from one region and refer to repetitive journals. Moreover, among the references there is a reference to a master degree thesis, which is probably an unpublished position.

The discussion in the article cannot be considered as detailed. The discussion (lines 569-756) confronts the research results with only 5 literature items. The discussion should be redrafted and supplemented. The Authors do not discuss their results with the quoted positions [28, 29], which refer to the analysed area.

Conclusions (lines 757-791) should also be thoroughly redrafted. The Authors should draw conclusions of global importance based on their research. In this form, the article has local significance and will not be of interest to a wide audience.

Minor comments:

There are many errors in the manuscript and redundant information (not all, just examples):

- lines 97-98: put forward six major functions such as raw material supply, carrying and transportation. (The Authors only mention 3 functions, not 6).

- line 136: Therefore, this research is based on the above four questions. (After this sentence, the Authors do not pose research questions (hypotheses), but reveal the research assumptions).

- lines 193-194: The data required for this study mainly include land use data, climate data, soil data, topographic data, remote sensing image data, and socioeconomic statistics. (Then no data that falls under the other categories is listed. The word 'mainly' is therefore completely incorrect and redundant.)

- lines 162-163: The rapid development of urbanization is usually accompanied by rapid changes in urban land use [29]. (This sentence concerns a commonly known fact, it is redundant and additionally does not require reference to the literature. Facts of this type commonly known to the scientific community are mentioned in the article many times).

- lines 740-742: Different from the research method that uses administrative divisions as the research unit, the research accuracy on the grid scale is stronger. And it better reflects the differences within the region, between the city and the country (This statement is intuitive and not revealing).

Add missing reference to using data and:

- lines 324-325: According to the Swedish carbon tax method in the "Forest Ecosystem 324 Service Function Evaluation Specification", this study takes PC=1.2 yuan/kg.

- lines 175-176: while there were few urbanization studies in the western region.

The editorial form of the manuscript submitted for review is highly unsatisfactory.

Editing comments:

Authors should pay attention to: correct quotation of the references in the text of the manuscript, standardization of the units of measure used in the manuscript (e.g. km or hm, square kilometres or km2 appear in the text), correct insertion of figures in the text and improvement of their quality. Improve the readability of the figures, instead of the names on the maps, enter the numerical markings, as in the case of the Districts: Xincheng, Beilin and Lianhu. I suggest also combining Fig. 1 and Fig. 2 into one figure. The use of the signs 'a, b and c' in figures (see Figs. 4 - 9) and the lack of reference to them in the text is also very puzzling. There is also no reference to Fig. 3 and Fig. 8 in the manuscript text. Additionally, I suggest that you rethink the use of the word 'scholar' in the manuscript in favour of the word 'researcher'. Consider introducing abbreviations in the manuscript, for example for a land system. You mention the research area excessively (the word Xi'an about 53 times).

Conclusions:

The Authors rightly note that multi-functional trade-offs and synergy research on land systems are important problems from point of view of geography and land science researchers. The design of the research is correct, however the manuscript should be thoroughly redrafted. Redundant information should be removed. Authors should also focus on highlighting the strengths of their research compared to other existing ones. However, when formulating conclusions, the Authors must rise above their area of ​​research.

Author Response

Dear  Reviewer 1,

Thank you for your comments concerning our manuscript entitled “Multi-function Tradeoffs of Land System in Urbanized Areas——a case study of Xi’an, China” (ID: land-1216239). Those comments are all valuable and very helpful for revising and improving our paper, as well as the important guiding significance to our researches. We have studied comments carefully and have made correction which we hope meet with approval. Revised portion are marked in red in the paper. The main corrections in the paper and the responds to the reviewer’s comments are as flowing(pdf file) :

Reviewer 2 Report

It is an interesting study of interaction between three ecosystem services that are provided by soils in an urbanized landscape. It is written to a good level and it contains all the essentials of a quality scientific study. It is also implemented by quality images.
Besides production functions, soils are also fulfilling ecological functions in the land ecosystems. Therefore it is necessary to take into account a whole range of parameters when assessing land use. (An example can be publication in the Journal of Maps: Integrated index of agricultural soil quality in Slovakia, or in the Pedosphere journal: Soil environmental index for Slovak agricultural land).

Comments:
• The aim of the study should be clearly explained in the introduction. The results of the study were to be expected. Nevertheless, they are interesting mainly at the national level.
• The names of some images (lines 191, 442, 478, 542) should be just below the images and should explain what the unit of measurement is.
• Table 5 is unclear. What is the significance of these values? Unit of measure is missing. The years are given in the column: Value of the environmental services function.
• The data in Table 6 are unclear. Where does the value 0.156 belong and why is the plus when the others are minus?
• The comparison of the results obtained with other studies in the discussion must be at a better level. It contains only general opinions, resp. results that are more appropriate for the conclusions.
• The conclusions are not sufficiently developed. The conclusion must contain a summary of the objectives, concrete results achieved and a proposal for their implementation.

Author Response

Dear Reviewer  2

Thank you for your  comments concerning our manuscript entitled “Multi-function Tradeoffs of Land System in Urbanized Areas——a case study of Xi’an, China” (ID: land-1216239). Those comments are all valuable and very helpful for revising and improving our paper, as well as the important guiding significance to our researches. We have studied comments carefully and have made correction which we hope meet with approval. Revised portion are marked in red in the paper. The main corrections in the paper and the responds to the reviewer’s comments are as flowing(pdf file):

Reviewer 3 Report

Some of the language structure is convoluted and lengthy and therefore difficult to comprehend. See, for example, lines 122 to 135. This is one long, continuous sentence that should be broken up into more coherent and shorter sentences. 

Figures 1 and 2 need to be re-set into the correct position. 

Author Response

Dear  Reviewer 3

Thank you for your comments concerning our manuscript entitled “Multi-function Tradeoffs of Land System in Urbanized Areas——a case study of Xi’an, China” (ID: land-1216239). Those comments are all valuable and very helpful for revising and improving our paper, as well as the important guiding significance to our researches. We have studied comments carefully and have made correction which we hope meet with approval. Revised portion are marked in red in the paper. The main corrections in the paper and the responds to the reviewer’s comments are as flowing(pdf file):

Round 2

Reviewer 1 Report

Thanks to the Authors of the manuscript for the answers to my questions and correction applied after the suggestions of all Rewievers. 
I am satisfied with most of these changes and responses.

Minor coments:
Fig. 1 is currently missing from the manuscript text. Probably this is due to a mistake in editing the text. Add Fig. 1.

Descriptions in some of the figures (eg. Fig. 5 and 6) are still illegible.

Author Response

Dear Editor and Reviewer 1

Thank you for your letter and for the reviewer 1’s comments concerning our manuscript entitled “Multi-function Tradeoffs of Land System in Urbanized Areas——a case study of Xi’an, China” (ID: land-1216239). Your second round of comments gave me a lot of encouragement. We really appreciate you for your carefulness and conscientiousness.  We have carefully made correction which we hope meet with approval. Revised portion are marked in red in the paper. The corrections in the paper and the responds to the reviewer 1’s comments are as flowing:(Please see the attachment)
